# Association between Immunologic Markers and Cirrhosis in Individuals from a Prospective Chronic Hepatitis C Cohort

**DOI:** 10.3390/cancers14215280

**Published:** 2022-10-27

**Authors:** Ilona Argirion, Jalen Brown, Sarah Jackson, Ruth M. Pfeiffer, Tram Kim Lam, Thomas R. O’Brien, Kelly J. Yu, Katherine A. McGlynn, Jessica L. Petrick, Ligia A. Pinto, Chien-Jen Chen, Allan Hildesheim, Hwai-I Yang, Mei-Hsuan Lee, Jill Koshiol

**Affiliations:** 1Division of Cancer Epidemiology and Genetics, National Cancer Institute, Rockville, MD 20850, USA; 2Division of Cancer Control and Population Sciences, National Cancer Institute, Rockville, MD 20814, USA; 3Slone Epidemiology Center, Boston University, Boston, MA 02215, USA; 4HPV Immunology Laboratory, Frederick National Laboratory for Cancer Research, Leidos, Biomedical Research, Inc., Frederick, MD 21702, USA; 5Genomics Research Center, Academia Sinica, Taipei 11529, Taiwan; 6Graduate Institute of Epidemiology and Preventative Medicine, College of Public Health, National Taiwan University, Taipei 10617, Taiwan; 7Institute of Clinical Medicine, National Yang-Ming University, Taipei 11221, Taiwan; 8Graduate Institute of Medicine, College of Medicine, Kaohsiung Medical University, Kaohsiung 80708, Taiwan; 9Biomedical Translation Research Center, Academia Sinica, Taipei 11529, Taiwan

**Keywords:** cirrhosis, hepatitis C, circulating immunologic proteins, inflammation, prospective studies

## Abstract

**Simple Summary:**

To date: no prospective cohort studies have been used to investigate the role of circulating immunologic markers as they relate to the progression of liver disease in those chronically infected with hepatitis C (HCV). Using data/samples from a prospective cohort of chronically HCV-infected individuals, we sampled 68 individuals who developed cirrhosis, 91 controls who did not develop cirrhosis, and 94 individuals who developed hepatocellular carcinoma (HCC). Using baseline plasma, we examined levels of 102 markers in individuals who developed cirrhosis vs. controls and those who developed HCC vs. cirrhosis. The findings from this study highlight the important role of immunological markers in predicting HCV-related cirrhosis.

**Abstract:**

Background: Chronic hepatitis C virus (HCV) infection can affect immune response and inflammatory pathways, leading to severe liver diseases such as cirrhosis and hepatocellular carcinoma (HCC). Methods: In a prospective cohort of chronically HCV-infected individuals, we sampled 68 individuals who developed cirrhosis, 91 controls who did not develop cirrhosis, and 94 individuals who developed HCC. Unconditional odds ratios (ORs) from polytomous logistic regression models and canonical discriminant analyses (CDAs) were used to compare categorical (C) baseline plasma levels for 102 markers in individuals who developed cirrhosis vs. controls and those who developed HCC vs. cirrhosis. Leave-one-out cross validation was used to produce receiver operating characteristic curves to assess predictive ability of markers. Lastly, biological pathways were assessed in association with cirrhotic development compared to controls. Results: After multivariable adjustment, DEFA-1 (OR: C2v.C1 = 7.73; *p* < 0.0001), ITGAM (OR: C2v.C1 = 4.03; *p* = 0.0002), SCF (OR: C4v.C1 = 0.19; *p*-trend = 0.0001), and CCL11 (OR: C4v.C1 = 0.31; *p*-trend= 0.002) were all associated with development of cirrhosis compared to controls; these markers, together with clinical/demographics variables, improved prediction of cirrhosis from 55.7% (in clinical/demographic-only model) to 74.9% accuracy. A twelve-marker model based on CDA results further increased prediction of cirrhosis to 88.0%. While six biological pathways were found to be associated with cirrhosis, cell adhesion was the only pathway associated with cirrhosis after Bonferroni correction. In contrast to cirrhosis, DEFA-1 and ITGAM levels were inversely associated with HCC risk. Conclusions: Pending validation, these findings highlight the important role of immunological markers in predicting HCV-related cirrhosis even 11 years post-enrollment.

## 1. Introduction

Hepatitis C virus (HCV) affects 130–210 million people worldwide and an estimated 350,000 people die from an HCV-related diseases every year [1]. While acute infection can resolve spontaneously, approximately 80% of HCV-positive individuals will transition from having an acute to a chronic infection, putting them at high risk for liver disease; in particular, liver cirrhosis affects 10–20% of chronically infected HCV individuals [2]. Furthermore, an estimated 10% of cirrhotic patients may go on to develop hepatocellular carcinoma (HCC) within 5 years [3]. The 2017 WHO Global Hepatitis report estimates incidence of HCV infection to be 6.0/100,000 in the Western Pacific Region and 14.8/100,000 in South East Asian region [4]. Given the significant impact of chronic HCV infection on both morbidity and mortality, it is important to improve our understanding of the underlying biology of HCV-related liver disease. 

HCV entry into the hepatocytes is believed to occur through a highly coordinated and complex effort involving several cell surface molecules [5]. Once in the host cell, HCV can directly modulate signaling and metabolic pathways by viral proteins, as well as indirectly induce host antiviral immune response, resulting in cell death [6,7]. Hepatic stellate cells (HSCs) appear primarily implicated in the fibrogenesis leading to liver cirrhosis and HCC [2]. HCV activates HSCs through the production of viral proteins that directly and indirectly stimulate HSCs to develop into abnormally overgrown myofibroblasts that do not degrade normally, causing scarring of the liver and eventually cirrhosis [2]. The release of microRNAs, interleukins (ILs), chemokines, inflammatory factors, neuroendocrine signals and oxidative stress markers regulate this crossover of HSCs to myofibroblasts [2,8,9]. While the macrocellular processes involved in cirrhosis have been the focus of several studies, many of the microcellular and immunological components regulating progression from chronic HCV infection to cirrhosis or from cirrhosis to HCC remain to be elucidated. 

Previous studies investigating specific immunological markers of HCV-related liver disease have identified a few important proteins [10,11,12]. Our group recently identified 11 immunological markers found to be associated with HCC when compared to non-cirrhotic controls in a cohort of chronically HCV-infected participants; these markers included ICAM-1, CXCL10, IL-8 and CES1 [10]. However, to our knowledge, no studies using samples from prospective cohorts have investigated the role of circulating immunologic markers as they relate to the progression of liver disease in those chronically infected with HCV [12,13]. To address this gap, we used the Risk Evaluation of Viral Load Elevation and Associated Liver Disease/Cancer-Hepatitis C Virus (REVEAL-HCV) cohort to identify associations between immunologic proteins and cirrhosis in comparison to chronically infected controls and HCC cases. 

## 2. Methods

### 2.1. Study Population

REVEAL-HCV comprises a longitudinal, community-based cohort of 1313 individuals who were seropositive for anti-HCV antibodies and enrolled (1991 through 1992) to investigate the long-term outcomes of HCV infection [10]. A subset of 1159 individuals were tested for HCV-RNA at baseline. Of these individuals, 789 participants were positive (677 of whom were HBsAg negative). REVEAL cohort members provided informed consent and underwent to high-resolution, real-time abdominal ultrasound and blood collection every 6–12 months through December 2008 [10]. This study was approved by the Institutional Review Boards of the College of Public Health of the National Taiwan University and National Yang-Ming University (Taipei, Taiwan).

In addition to abdominal ultrasound, data linkage with the National Taiwan Health Insurance Database was used to identify incident cirrhotic cases after 1999 [10]. Cirrhosis was assessed through 2010 and HCC was assessed through 2013. HCC cases were identified using abdominal ultrasound, α-fetoprotein testing or data linkage with the Taiwanese National Cancer Registry [10]. Further confirmation of cases occurred via data linkage with Taiwan’s national death certification system and histopathological examinations of HCC cases in medical review charts [10]. HCC was diagnosed based on identification of a lesion by ≥2 imaging techniques (angiogram, computed tomography, or ultrasonography) or a serum a-fetoprotein level ≥ 400 ng/m and 1 imaging technique [10,14,15]. 

For the current study, the cirrhotic group was comprised of 68 participants who developed cirrhosis but not HCC between their baseline visit and end of follow-up. The 91 chronically infected controls who did not develop cirrhosis were incidence-density matched to 94 HCC cases based on age (5-year age groups), sex, follow-up time (±1 year), previous sample thaw, and viral load (closest viral load to the corresponding HCC case); three controls were matched to two HCC cases each. 

### 2.2. Marker Measurements

250 µL tubes of heparin plasma were stored at −70 °C at the Academia Sinica in Taipei, Taiwan upon collection. For the purposes of this study, samples collected at or as close to baseline as possible were selected and aliquoted into 96 well plates and shipped to the United States National Cancer Institute. These samples were then evaluated for 184 immunologic proteins using the Olink (Uppsala, Sweden) inflammation and cardiometabolic panels, requiring 1 µL of sample per panel. The inflammation panel includes cytokines, chemokines and other inflammatory markers identified in collaboration with experts and believed to be related to cancer. The cardiometabolic panel was used for this study due to its large number of liver-specific proteins. For quality control (QC) purposes, we included an additional 24 blinded duplicates. Several markers were excluded for the following reasons: undetectable in >90% of samples (N = 12), intraclass correlation coefficients (ICCs) less than 80% (N = 69), coefficient of variation (CV) greater than 25% (N = 1). The final number of markers included in the analysis was 102 (Appendix A). 

### 2.3. Statistical Analysis

Serological markers were grouped and analyzed as categorical (C) variables in the following manner: among markers with values above the lowest limit of detection (LLOD) in ≥75% of controls (N = 83), four categories were constructed based on values above the LLOD, with values at or below the LLOD additionally included in the lowest category. For those markers with values above the LLOD in 50–75% of controls (N = 2), four groups were created, with those values at or below the LLOD comprising the lowest category, and values above the LLOD divided into three categories. For those markers with values above the LLOD in 25–50% of controls (N = 4), three groups were established, with the lowest group comprised of values at or below the LLOD and the remaining values dichotomized around the median. Lastly, for those markers with values above the LLOD in <25% of controls (N = 13), a binary variable was created based on values at/below vs. above LLOD. We used unconditional polytomous logistic regression models to estimated odds ratios (ORs) for the association between categorical circulating immunologic markers and stages of liver disease; to do this, individuals who developed cirrhosis were compared to chronically infected controls, and individuals who developed HCC were compared to those who developed cirrhosis. We chose covariates based on a priori knowledge and backward selection. The finalized models were adjusted for continuous age, sex, continuous years of follow-up (time from baseline/entry to HCC diagnosis, death, or end of study [31 December 2011], whichever came first), ALT level (<15, 15–44, ≥45 U/L), alcohol use (yes/no) and smoking status (ever/never). To account for multiple comparisons, models were corrected using a Benjamini-Hochberg false discovery rate [16] of 10%. 

To assess the role of temporality, we conducted an analysis stratified by median time to cirrhosis diagnosis among cirrhotic participants (10.75 years). Finally, due to the large number of HCC participants that had underlying cirrhosis (N = 69), we conducted an analysis stratified by the presence or absence of underlying cirrhosis. Statistical significance for these sensitivity analyses was determined using an α = 0.05. 

In addition to assessing individual markers, we used canonical discriminant analysis (CDA) to identify linear combinations of immune markers that could discern between the same comparison groups evaluated in the primary analysis using backward selection (α = 0.05). To evaluate and compare the predictive ability of biomarker inclusion for cirrhosis vs. controls and HCC vs. cirrhosis, we used leave-one-out cross validation [17] to calculate the area under the receiver operating characteristic curves (AUCs) for the following three models: (i) clinical/demographic characteristics alone (age, sex, years of follow-up, ALT, drinking and smoking), (ii) clinical/demographic variables plus the markers found to be significant in the logistic regression model, and (iii) clinical/demographic variables plus the markers found to be significant in the CDA. Statistical significance was determined based on the DeLong, DeLong, and Clarke-Pearson method [18] using an α = 0.05.

Lastly, to assess immunological markers by biological pathway, we leveraged Olink’s ‘Biological Process’ resources to group markers for analyses. To limit the burden of multiple testing, we selected pathways believed to be relevant to our outcomes a priori and excluded those including fewer than ten makers contributed (after QC); this analysis included ten pathways (Appendix A). Continuous measures of the markers were used for this analysis, and markers that did not pass QC, and values of other markers that were below the LLOD, were excluded. The ARTP package was used to obtain pathway-specific summary *p*-values for associations of each pathway and cirrhosis (compared to controls), after adjustment for age, sex, years of follow-up, ALT level, drinking and smoking [19]. In brief, the ARTP analysis consists of the following steps: (i) fit linear regression models separately for the continuous values of each marker as the outcome and cirrhosis/HCC/control status as the independent variable, (ii) combine the *p*-values for the regression coefficient associated with cirrhosis/HCC/control status for all markers in a particular pathway using an adaptive rank truncated product statistic to obtain a pathway-specific *p*-value, (iii) evaluate the statistical significance of the pathway-level *p*-value (i.e., the test statistic) using a highly efficient permutation algorithm obtained by permuting case–controls status in the regression models and combining the permutation *p*-values of all markers in the same pathway to yield a *p*-value for association of the pathway with cases-control status. While the pathways were chosen based on a priori knowledge, we also applied a Bonferroni correction to the *p*-values when interpreting our results. 

The pathway analysis was conducted in R (version 4.0.3 [2020–10–10]).

All other statistical analyses were performed with SAS software version 9.4 (SAS Institute, Inc., Cary, NC, USA). 

## 3. Results 

### 3.1. Study Characteristics

The median time from sample collection to cirrhosis diagnosis was 10.75 years (range: 0.4–19.0); the median time from sample collection to HCC diagnosis was 13.3 years (range: 2.3–21.6). Median follow-up for those who developed cirrhosis was 19.9 years, (range: 7.5–20.9), for the HCC cases was 15.3 years (range: 3.6–20.8) and for controls was 19.9 years (range: 2.1–20.8). Males comprised 57.5%, 51.5% and 57.0% of HCC cases, cirrhotic participants, and controls, respectively. HCC participants were more likely to report smoking (42.5%) when compared to cirrhotic participants (25%) and controls (29.7%). HCC cases also were more likely to have higher baseline RNA positivity (91.5%) compared to cirrhotic participants (75.0%). A higher number of HCC individuals presented with ALT levels ≥ 45 U/L (36.2%) compared to cirrhotic participants (19.1%) and controls (19.8%). There were no significant differences in age, BMI, alcohol use or family history of HCC between persons with cirrhosis, HCC and controls (Table 1). 

### 3.2. Associations with Cirrhosis vs. Controls

Analysis of individual markers yielded four proteins that were found to be positively associated with cirrhosis development compared to controls (Table 2; Appendix A). Cirrhotic participants were seven times more likely to exhibit higher levels of defensin alpha 1 (DEFA-1) at baseline compared to controls OR_C2v.C1_ = 7.7 (95% confidence interval [CI]: 3.0–20.2) and four times more likely to exhibit higher levels of integrin subunit alpha M (ITGAM) OR_C2v.C1_ = 4.0 (95% CI: 1.9–8.4). Conversely, stem cell factor (SCF) was found to be inversely associated with cirrhosis development [OR_C4v.C1_ = 0.19 (95%CI: 0.07–0.53)], as was C-C motif chemokine ligand 11 (CCL11) [OR_C4v.C1_ = 0.31 (95%CI: 0.12–0.80)].

In a sensitivity analysis aimed at assessing temporality on the observed associations, we found the magnitude of the associations of DEFA-1, ITGAM and SCF with cirrhosis development to be notably stronger among those who developed cirrhosis early (<10.75 years after sample collection) compared to those who developed cirrhosis late (≥10.75 years after sample collection) (Figure 1). Among cirrhotic individuals diagnosed late in study follow-up (≥10.75 years after sample collection) DEFA-1, ITGAM and CCL11 were still significantly associated with cirrhotic development when compared to controls. However, the magnitude of association with late cirrhosis development was attenuated for DEFA-1 and ITGAM. 

CDA indicated 12 markers implicated in the progression of cirrhosis vs. controls, including DEFA-1, artemin (ARTN), regenerating family member 1 alpha (REG1A), three c-c motif chemokines (CCL3,CCL4, CCL18), cluster of differentiation 40 (CD40), C-X-C motif chemokine ligand 10 (CXCL10), extracellular newly identified RAGE-binding protein (EN-RAGE), latency-associated peptide transforming growth factor beta-1 (LAP/TGF-beta-1), human monocyte chemoattractant protein (MCP4), and programmed death-ligand 1 (PD-L1). When compared to the clinical/demographic ROC model, which yielded an AUC of 55.7% (95%CI: 46.3–65.1%), the addition of the four markers isolated through the unconditional polytomous regression models (DEFA-1, ITGAM, SCF and CCL11) increased the AUC to 74.9% (95%CI: 67.2–82.5%). The 12-marker linear combinations from the CDA in addition to the clinical/demographic variables yielded an AUC of 88.0% (95%CI: 82.8–93.1%) (Figure 2). The ROC contrast test revealed that the addition of these 12 biomarkers significantly improved predictability of cirrhosis development compared to the clinical/demographic only model (*p*-value < 0.0001).

Six molecular processes in the pathway analysis were significantly associated with cirrhosis at the nominal 0.05 significance level, including: cell adhesion (*p*-value = 0.001), cellular response to cytokine stimulus (*p*-value = 0.02), extracellular matrix organization (*p*-value = 0.02), inflammatory response (*p*-value = 0.01), complement activation (*p*-value = 0.02), and immune response (*p*-value = 0.02) (Figure 3). Only the cell adhesion pathway remained significant after Bonferroni correction. Two biomarkers in this pathway, CCL4 and LAP-TGF-beta-1, also improved predictability of cirrhosis in the 12-marker CDA model. DEFA-1 appeared in the immune response pathway. ITGAM appeared in cell adhesion, extracellular matrix organization, complement activation, and immune response pathways. SCF appeared in the cell activation pathways and CCL11 appeared in the cell adhesion, cellular response to cytokine stimulus, and inflammatory response pathways. 

### 3.3. Associations with HCC vs. Cirrhosis

Regarding HCC risk, development of HCC showed inverse associations compared to persons with cirrhosis with two proteins, including DEFA-1 [OR_C2v.C1_ = 0.06 (95% CI: 0.02–0.20)] and ITGAM [OR_C2v.C1_ = 0.20 (95% CI: 0.08–0.40)]. (Appendix A). Sensitivity analyses stratifying persons with HCC by the presence of underlying cirrhosis was conducted in order to further elucidate these associations. Effect sizes were similar for HCC cases without underlying cirrhosis (N = 25) and HCC cases with underlying cirrhosis (N = 69) (Appendix A). 

CDA found 14 markers that differentiate individuals with HCC from individuals with cirrhosis, including CCL18, DEFA1, ITGAM, ARTN, REG1A, CD40, EN-RAGE, LAP/TGF-B1, PDL1, as well as serpin family a member 7 (SERPINA 7), caspase 8 (CASP 8), fibroblast growth factor 23 (FGF-23), interleukin-7 (IL-7), and vascular endothelial growth factor A (VEGFA). The ROC clinical/demographic model produced an AUC of 72.5% (95%CI: 64.6–80.4%). The addition of the 4 markers from the unconditional polytomous regression models yielded an increased AUC of 84.9% (95%CI: 79.0–90.8%), and the 14-marker model including covariates yielded an AUC of 95.4% (95%CI: 92.5–98.2%) (Figure 2). The AUC of the model using the 14 biomarkers was significantly higher compared to the clinical/demographic only model (*p*-value < 0.0001).

## 4. Discussion

This is the first study, to our knowledge, to assess the association between a wide range of immunologic proteins and cirrhosis development in a prospective cohort of individuals chronically infected with HCV. We identified two individual proteins (DEFA-1, and ITGAM) that were positively associated, and two individual proteins (CCL11 and SCF) that were negatively associated with the development of cirrhosis. While these associations appeared particularly strong among those diagnosed within eleven years of sample collection, DEFA-1, CCL11 and ITGAM proved to be robust predictors of cirrhotic development even among those diagnosed eleven years after sample collection. The addition of these four proteins significantly improved model predictability of cirrhosis by approximately 19% (55.7% vs. 74.9%) compared to a model with clinical/demographic variables alone. CDA, that accounts for marker correlations, identified 12 markers that in addition to the clinical/demographic model predicted development of cirrhosis with 88% accuracy in a cross-validation model. DEFA-1 and ITGAM and were inversely associated with HCC when compared to cirrhotic participants, potentially suggesting a bi-directional effect of these markers during the natural progression of disease among HCV infected individuals. Lastly, we identified the cell adhesion pathway as a biological key pathway in fibrotic progression. This pathway includes LAP-TGF-beta-1, a protein that plays a key role in the progression of liver fibrosis by preventing hepatocyte regeneration [20].

Eotaxin (CCL11) is a chemotactic cytokine produced by epithelial cells, endothelial cells, fibroblasts, and monocytes [21] and has previously been reported to be associated with eosinophil-associated gastrointestinal diseases, allergic asthma, pulmonary fibrosis and atherosclerosis [22,23,24,25,26]. While few studies have investigated the role of CCL11 in liver diseases, eotaxin gene expression has been shown to be up-regulated in senescent compared to proliferative HSCs [27]. A study by Landi et al. compared patients with chronic HCV to controls and found elevated serum CCL11 levels in those with primary sclerosing cholangitis but reduced levels in those with primary biliary cirrhosis and autoimmune hepatitis [28]. Our study similarly found an inverse association between CCL11 and cirrhosis development, which may be explained by the repeated liver injury sustained through HCV infection, resulting in greater cellular proliferation and thus reduced CCL11 expression. 

Of the chemokines contributing to improved predictability of cirrhosis, CXCL10, CCL3, and CCL4 have been implicated in progression of liver disease. HCV-induced CXCL10 has been shown to increase hepatocyte turnover, resulting in the development of fibrosis and cirrhosis [29] and has also been implicated as a positive predictor of liver fibrosis recurrence in HCV-infected liver transplant patients [30]. Similarly, CCL3 and CCL4 have been shown to interact with CCR5, a chemokine receptor implicated in hepatic inflammation [31]. All three of these chemokines, in addition to several others (Appendix A), appeared in the cellular response to cytokine stimulus pathway, further supporting the importance of inflammatory processes in liver disease. 

Working synergistically, beta-chemokines (i.e., CC chemokines) have been shown to induce neutrophils to release alpha-defensins [32]. DEFA-1, otherwise known as HNP-1, is a type of defensin known for inhibiting viral entry and replication, recruiting macrophages, and destroying viral particles through chemotactic, immunomodulatory, and cytotoxic activity [33,34,35,36]. DEFA-1 appeared as a biomarker in the immune response pathway analysis. At least two studies have implicated HNPs 1-4 in the progression of liver fibrosis in patients with chronic hepatitis C [37,38]. The likely mechanism of action involves stimulation of liver HSCs and Kupffer cells, which under the influence of DEFA-1 release pro-inflammatory cytokines, IL-8 and TNF-alpha, contribute to the maintenance of inflammation [2,39,40]. The inverse association between levels of DEFA-1 and HCC highlights a potential difference in the role of this marker in the progression of cancer. HNPs 1-3, have been linked to blockage of angiogenesis via direct inhibition of vascular endothelial growth factor (VEGF), induction of cancer cell apoptosis, as well as tumor growth inhibition—prohibiting carcinogenesis [41,42,43,44,45,46]. Consequently, DEFA-1 may have a bi-directional effect in the progression of liver disease in the context of chronic HCV infection. 

ITGAM is an important protein linked to the phagocytosis of particles and mediation of inflammation via leukocyte adhesion [47,48]. The protein serves as a subunit for α_M_β_2_, a phagocytic receptor found on numerous cells of the humoral immune system including macrophages, neutrophils, and natural killer cells [47,49,50]. Integrins have been shown to co-localize with TGF-β, together influencing both fibrogenesis and carcinoma development [51]. Elevated ITGAM levels have been observed in peripheral blood leukocytes of individuals with liver cirrhosis [52]. It is plausible that the positive association between ITGAM levels and cirrhosis relates to the role of integrins in the contraction of myofibroblasts [51]. Hepatic stellate cells transition to myofibroblasts following increased intracellular levels of calcium and ROS; it is likely that integrins influence myofibroblast expansion after this step [2]. LAP/TGF- β, a marker found to improve predictability of cirrhosis in the CDA, differentiates myofibroblasts and suppresses fibrogenesis when inhibited [53,54]. Interestingly, when compared to cirrhotic participants, those that developed HCC were less likely to have elevated levels of ITGAM. While the role of ITGAM in the context of HCC development remains to be fully elucidated, a recent study found that the inhibition of ITGAM prevents expression of microRNA *Let7a*, consequently inducing cMyc expression that results in vascular maturation, immune suppressive macrophage polarization, and enhanced tumor growth [50]. Thus, this is consistent with the inverse association that we observe. 

Several proteins appeared in the cell adhesion pathway of our pathway analysis which remained significantly associated with cirrhosis development even after Bonferroni corrections. Cell adhesion molecules (CAMs), such as integrins and selectins, activate and recruit leukocytes by migrating these cells from the vascular endothelium to inflamed liver tissue to trigger a cytotoxic response [20]. Of note, ITGAM, LAP-TGF-beta-1, PD-L1 and CCL18 contribute to cell adhesion. All four of these proteins contributed significantly to prediction of cirrhosis in CDA. PD-L1 likely contributes to immunosuppression by binding to its receptor PD-1 on cytotoxic T-cells to inhibit apoptosis as well as cellular proliferation [55]. Several chemokines in this pathway have been shown to impact human HSC proliferation and migration (e.g., CCL4) [56]. The strong association between the cell adhesion pathway and cirrhosis development likely reflects the number of CAMs present in cirrhotic individuals within our cohort.

Lastly, stem cell factor (SCF) is important in cellular proliferation and has been reported to have anti-apoptotic properties. Characterized as a hematopoietic factor, SCF induces stem cell maturation and differentiation [57,58]. A few recent reports have highlighted the role of SCF in tissue remodeling, including liver regeneration following injury or after partial hepatectomy [59,60,61,62,63]. Several of these studies have demonstrated detectable levels of c-Kit, the receptor for SCF, during liver damage and repair processes. Given these reported associations, our inverse findings are somewhat surprising. Nevertheless, SCF exists in both membrane-bound and soluble forms, and the relative importance of each remains poorly understood [63]. The current literature has largely focused on the presence of SCF and c-Kit mRNA in liver tissue through the utility of murine models. One recent study of serum SCF levels found significantly lower levels of SCF in fulminant hepatitis patients when compared to acute hepatitis patients and controls, and that these decreased levels were associated with poorer prognosis; the authors of this manuscript hypothesized that impaired SCF production in oval cells and bone marrow cells may be associated with poor regeneration of liver cells [64]. In our study, we found that when stratified by time to diagnosis, the association between SCF and cirrhosis was only significant among those who developed cirrhosis early (<10.75 years after sample collection). These results may similarly demonstrate a reduced capability of tissue repair among patients, resulting in quicker progression to cirrhosis. Further studies need to be conducted to better understand the role of circulating SCF in liver disease.

A major strength of this study is that it is based in the REVEAL-HCV cohort, a large, well-established prospective cohort that allows the opportunity to investigate differences in immunological markers among chronically infected HCV individuals. However, the current study does have limitations. We were unable to identify cirrhotic cases before 1999 through data linkage, thus there may be underdiagnosis of cases between 1991 and 1999. However, abdominal ultrasonography, a reliable diagnostic technique for cirrhosis, occurred at consistent biannual or annual periods throughout the study. Several studies have demonstrated the reliability of ultrasound to detect cirrhosis [65,66]; in the REVEAL cohort, a medical record review supported the reliability in identifying cirrhosis, with a kappa of 0.81. It is unlikely that an aberrant increase in underdiagnosis would result from a lack of data linkage during this period. In addition, we did not have information available on anti-HCV therapies. Nevertheless, the Taiwanese universal health care system through 2003 did not reimburse patients for antiviral therapies. After 2003, only patients under high-risk conditions and strict criteria were reimbursed. Consequently, it is reasonable to believe that patients did not receive antiviral therapies at the time of the study. Given recent reports of improved immune system-related markers in patients following direct-acting antiviral (DAA) therapy [67], future studies should be conducted to evaluate the associations between immunologic markers and cirrhosis development considering DAA treatment and viral clearance. 

## 5. Conclusions

Describing the natural history of disease in high-risk patients infected with chronic HCV can translate to improved preventative and therapeutic applications for liver disease. In this study evaluating the role of an extensive list of circulating immunologic markers, we identified several proteins associated with cirrhotic development, most of which remained predictive even 11 years after study entry. These results highlight the important biological role of immune and inflammatory markers DEFA-1, ITGAM, SCF, CCL11 as well as the cell adhesion pathway in liver disease progression. The strength of these associations suggests that the inclusion of serum biomarkers may improve risk assessment for prognosis of individuals with chronic HCV infection. Although replication and validation are needed, these findings elucidated important immunological pathways that exist in HCV-mediated liver disease.

## Figures and Tables

**Figure 1 cancers-14-05280-f001:**
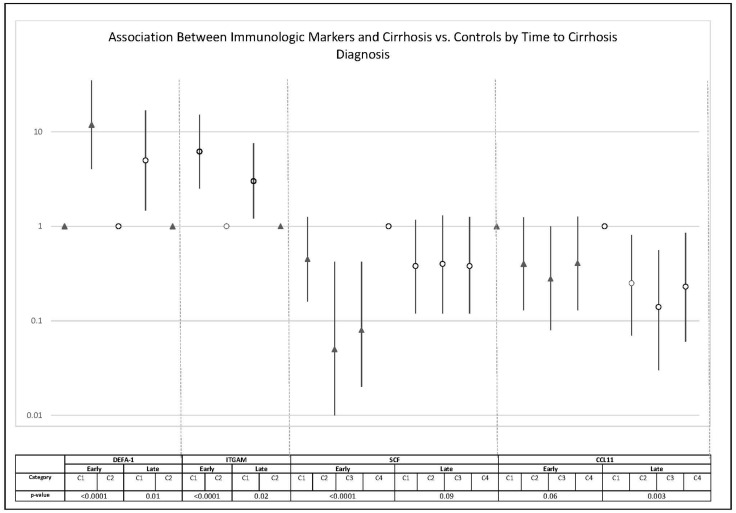
REVEAL-HCV odds ratios* and 95%CIs stratified by time to cirrhosis diagnosis in those who developed cirrhosis vs. controls. * Adjusted for age, sex, years of follow-up, serum alanine aminotransferase (ALT) level, alcohol, and smoking.

**Figure 2 cancers-14-05280-f002:**
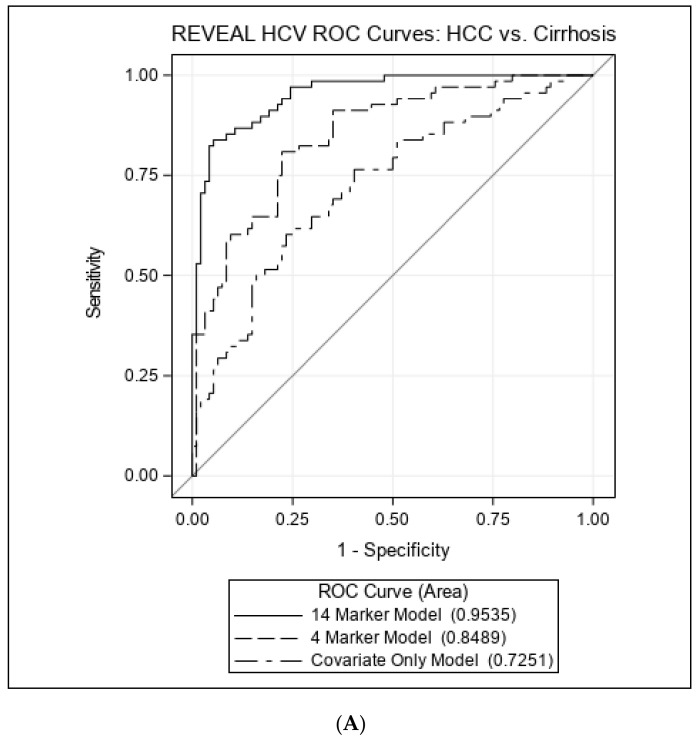
Receiver operating characteristic (ROC) curves comparing predictive models. (**A**) Comparison between cirrhotic participants and controls. Twelve marker model: CCL18, DEFA1, REG1A, ARTN, CCL3, CCL4, CD40, CXCL10, EN-RAGE, LAPTGF-B1, MCP-4, PD-L1 + covariates. Four marker model: DEFA-1, ITGAM, CCL11, SCF + covariates. Covariate Only Model: age, sex, years of follow-up, serum alanine aminotransferase (ALT) level, alcohol, and smoking (**B**) Comparison between HCC and cirrhotic participants. Fourteen marker model: CCL18, DEFA1, ITGAM, REG1A, SERPINA7, ARTN, CASP-8, CD40, EN-RAGE, FGF-23, IL7, LAPTGF-B1, PDL1, VEGFA + covariates. Four marker model: DEFA-1, ITGAM, CCL11, SCF + covariates. Covariate Only Model: age, sex, years of follow-up, serum alanine aminotransferase (ALT) level, alcohol, and smoking.

**Figure 3 cancers-14-05280-f003:**
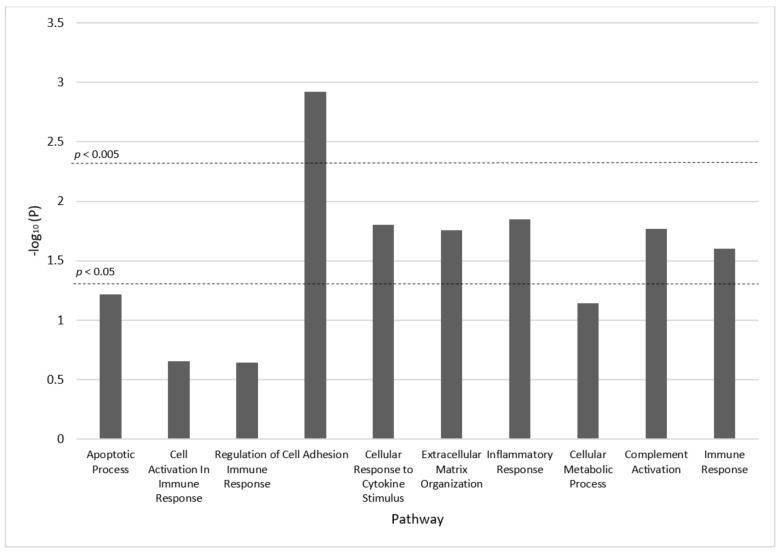
Pathway analysis results.

**Table 1 cancers-14-05280-t001:** Participant characteristics.

Characteristic	Cirrhosis Cases	HCC Cases	Controls
Total N	68	94	91 ^a^
% Male	51.5%	57.5%	57.1%
Median Age at Sample Date (Range)	54.0 (34–65)	55.5 (34–65)	56.0 (30–65)
Median Year of Serum Collection (Range)	1991 (1991–1992)	1993 (1992–1997)	1993 (1992–1996)
Median Years of Follow-up (Range) *	19.9 (7.5–20.9)	15.3 (3.6–20.8)	19.9 (2.1–20.8)
Median BMI (Range)	24.4 (17.9–32.3)	25.0 (18.4–37.6)	23.3 (18.6–33.7)
ALT, N (%)			
<15	19 (27.9)	20 (21.3)	31 (34.1)
15–44	36 (52.9)	40 (42.6)	42 (46.2)
≥45	13 (19.1)	34 (36.2)	18 (19.8)
Smoking			
No	51 (75.0)	54 (57.5)	64 (70.3)
Yes	17 (25.0)	40 (42.5)	27 (29.7)
Alcohol drinking			
No	63 (92.7)	85 (90.4)	82 (90.1)
Yes	5 (7.3)	9 (9.6)	9 (9.9)
Family history of HCC			
No	67 (98.5)	93 (98.9)	88 (96.7)
Yes	1 (1.5)	1 (1.1)	3 (3.3)
Baseline RNA positivity †			
No	14 (20.6)	8 (8.5)	8 (8.8)
Yes	51 (75.0)	86 (91.5)	83 (91.2)
Median HCV RNA level at sample date	17,400	40,400	25,700
Median time to event in years (Range)	10.75 (0.37–19.02)	13.3 (2.29–21.59)	-

^a^ Three controls were double matched to cases. * Time from baseline/entry to HCC diagnosis, death, or end of study (31 December 2011), whichever came first. † Numbers do not add to total due to missing values. Abbreviations: BMI, body mass index; ALT, alanine aminotransferease; HCC, hepatocellular carcinoma.

**Table 2 cancers-14-05280-t002:** Odds ratios * and 95% confidence intervals [Cis] for associations between selected markers and cirrhosis versus non-cirrhotic controls.

Analyte	OR (95% CI)			*p*-Value ^‡^	FDR-Corrected *p*-Value ^‡^
	C2 vs. C1	C3 vs. C1	C4 vs. C1		
Cirrhosis vs. Control					
DEFA-1	7.73 (2.96–20.17)	NA	NA	<0.0001	0.003
ITGAM	4.03 (1.94–8.36)	NA	NA	0.0002	0.006
SCF	0.40 (0.17–0.94)	0.19 (0.07–0.53)	0.19 (0.07–0.53)	0.0001	0.006
CCL11	0.30 (0.12–0.75)	0.20 (0.07–0.57)	0.31 (0.12–0.80)	0.002	0.05

* Adjusted for age, sex, years of follow-up, serum alanine aminotransferase (ALT) level, alcohol, and smoking. Abbreviations: OR, odds ratio; FDR, false discovery rate; NA, not applicable (marker only has two categories). ^‡^ *p*-values calculated as *p*-trend for analytes with more than two categories

## Data Availability

The analytic dataset used in this study are available upon request from the senior authors (meihlee@nycu.edu.tw or koshiolj@mail.nih.gov). The data are not publicly available because of privacy and/or ethical restrictions.

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
