# Peer review of "Association between Immunologic Markers and Cirrhosis in Individuals from a Prospective Chronic Hepatitis C Cohort"

_cancers, 2022, doi:10.3390/cancers14215280_

Round 1

Reviewer 1 Report

An interesting, investigative piece of research work. Representative cohort selection.  Carefully conducted research. . Clearly described methodology. Interesting presentation of the results on early and late cirrhosis. However, tables 3 and 5 contain too much information and are hence unreadable - perhaps the form of presentation is worth revisiting? Broad discussion, but rather general inference. Please reconsider the conclusions.

Author Response

Thank you for your comments and thoughtful review. To address this reviewer’s comments, we have done the following:

Table 3: We have removed table 3 and have instead presented the data as a figure (Figure 1). We hope this aids in clarifying the presentation of this information.

Table 5: We have split up supplemental table 5 into two separate tables: Table 5a (cirrhosis vs. controls) and Table 5b (HCC vs. cirrhosis).

Conclusions: We thank the reviewer for their comment. Given the sparse literature currently available on the role of circulating immunologic markers on cirrhosis progression in the context of HCV infection, we are hesitant to overstate our findings. Nevertheless, we have added the following to the conclusion: ‘These results highlight the important biological role of immune and inflammatory markers DEFA-1, ITGAM, SCF, CCL11 as well as the cell adhesion pathway in liver disease progression.’

In addition to the requested edits, we would also like to point out a few corrections that were made to the manuscript. An error was identified shortly after submission that altered the results of one of the analyses. In brief, the error was due to a merging issue with the p-values wherein the p-values from the HCC vs. cirrhosis analysis were applied to the cirrhosis vs. controls table, consequently, the results have now been updated omitting CCL18/ARTN and instead presenting CCL11 and SCF as the significant proteins associated with cirrhosis. Changes have been made throughout and the full updated output from the analysis can be found in supplemental table 5a. We apologize for this mistake and hope that our updated manuscript is satisfactory.

Reviewer 2 Report

The manuscript presents extremely important research results. Authors try to investigate the pathway of cirrhosis and/or HCC development. So far it is not well understood, particularly in HCV-related HCC and those without liver cirrhosis (that is quite rare, but may occur).

The methods and ptresentation of results are of high quality. The most important disadvantage of the study is mentioned in this part of manuscript discussing limitations of this study. REVEAL-HCV cohort is well-established prospective cohort, but assessing immunological markers without the history of therapy is extremely difficult with the risk of bias. Particularly, that probably IFN-based therapies were probably used in many patients. In our research that were already submitted we have found even relationship of immunologic markers to the type of DAA. So, I think if it is not possible to find out if any therapies were used in the cohort, it should more clearly included in discussion.

The second is the method of cirrhosis diagnosis. I am slightly surprised it was US-based diagnosis. Is it really the most reliable method? It is a very basic question when one try to explain the role of immune response in HCC in cirrhotic versus non-cirrhotic patients. The risk of misdiagnosis is the most important factor that may bias results and their interpretation.

My last comment is about Supplementary Table 3: "Markers tested in REVEAL-HBV"?? Is it another cohort or an error?

Author Response

Thank you for your thoughtful review and feedback. We have addressed the reviewer’s comments in the following manner:

Therapy: We have clarified in the limitations section of the discussion that ‘…we did not have information available on anti-HCV therapies”. Nevertheless, we do not believe treatment would greatly impact our results as Taiwanese universal health care system through 2003 did not reimburse patients for antiviral therapies. After 2003, only patients under high-risk conditions and strict criteria were reimbursed. While it would be wonderful to assess the impact for HCV treatment on the associations reported in this paper, we simply are unable to do so in this cohort. In light of this, we have added the following to the manuscript: “Given recent reports of improved immune system-related markers in patients following direct-acting antiviral (DAA) therapy [65], future studies should be conducted to evaluate the associations between immunologic markers and cirrhosis development considering DAA treatment and viral clearance.”

Cirrhosis diagnosis: Thank you for your comment. A study by Kelly et al. found the sensitivity of ultrasound to be 80% among patients with advanced fibrosis on liver biopsy (PMID: 30166950). Another study by Yen et al. found that using ultrasound-identified cirrhosis to predict compensated cirrhosis sensitivity was 34.0%, the specificity was 97.1%, the positive predictive value was 89.8%, the negative predictive value was 66.1% (PMID: 31277150). In our cohort, a medical record review supports the reliability of ultrasound for identifying cirrhosis. In personal correspondence with the cohort PI: “we compared cirrhosis status for subjects with medical chart review and record in national health insurance database. The kappa is 0.81. and there were 6/67 (8.9%) with disagreement of cirrhosis status.” To address this comment we added the following to the limitations paragraph: ‘Several studies have demonstrated the reliability of ultrasound to detect cirrhosis[65,66]; in the REVEAL cohort, a medical record review supported the reliability in identifying cirrhosis, with a kappa of 0.81.’

Supplemental table 3: Thank you for pointing out this typo, it has been corrected to read: ‘Markers tested in REVEAL-HCV; coefficients of variation (CVs) and intraclass correlation coefficients (ICCs) for markers with >90% detection among hepatocellular carcinoma (HCC) cases, cirrhosis cases, or non-cirrhosis controls; and reason for exclusion if excluded from analysis’

In addition to the requested edits, we would also like to point out a few corrections that were made to the manuscript. An error was identified shortly after submission that altered the results of one of the analyses. In brief, the error was due to a merging issue with the p-values wherein the p-values from the HCC vs. cirrhosis analysis were applied to the cirrhosis vs. controls table, consequently, the results have now been updated omitting CCL18/ARTN and instead presenting CCL11 and SCF as the significant proteins associated with cirrhosis. Changes have been made throughout and the full updated output from the analysis can be found in supplemental table 5a. We apologize for this mistake and hope that our updated manuscript is satisfactory

Round 2

Reviewer 2 Report

I have gone through the revised version of this manuscript. I think explanations and comments to my review are apropriate and I think the manuscript might be accepted in revised form.